# Microbial Consortia and Mixed Plastic Waste: Pangenomic Analysis Reveals Potential for Degradation of Multiple Plastic Types via Previously Identified PET Degrading Bacteria

**DOI:** 10.3390/ijms23105612

**Published:** 2022-05-17

**Authors:** Sabrina Edwards, Rosa León-Zayas, Riyaz Ditter, Helen Laster, Grace Sheehan, Oliver Anderson, Toby Beattie, Jay L. Mellies

**Affiliations:** 1Biology Department, Reed College, Portland, OR 97202, USA; sabrinaredwards@reed.edu (S.E.); ditterr@reed.edu (R.D.); lasterh@reed.edu (H.L.); tobeattie@reed.edu (T.B.); 2Biology Department, Willamette University, Salem, OR 97301, USA; rleonzayas@willamette.edu (R.L.-Z.); gasheehan@willamette.edu (G.S.); ofanderson@willamette.edu (O.A.)

**Keywords:** biodegradation, poly(ethylene)terephthalate (PET), polyhydroxyalkanoate (PHA), plasticizers, mixed plastics, pangenomes

## Abstract

The global utilization of single-use, non-biodegradable plastics, such as bottles made of polyethylene terephthalate (PET), has contributed to catastrophic levels of plastic pollution. Fortunately, microbial communities are adapting to assimilate plastic waste. Previously, our work showed a full consortium of five bacteria capable of synergistically degrading PET. Using omics approaches, we identified the key genes implicated in PET degradation within the consortium’s pangenome and transcriptome. This analysis led to the discovery of a novel PETase, EstB, which has been observed to hydrolyze the oligomer BHET and the polymer PET. Besides the genes implicated in PET degradation, many other biodegradation genes were discovered. Over 200 plastic and plasticizer degradation-related genes were discovered through the Plastic Microbial Biodegradation Database (PMBD). Diverse carbon source utilization was observed by a microbial community-based assay, which, paired with an abundant number of plastic- and plasticizer-degrading enzymes, indicates a promising possibility for mixed plastic degradation. Using RNAseq differential analysis, several genes were predicted to be involved in PET degradation, including aldehyde dehydrogenases and several classes of hydrolases. Active transcription of PET monomer metabolism was also observed, including the generation of polyhydroxyalkanoate (PHA)/polyhydroxybutyrate (PHB) biopolymers. These results present an exciting opportunity for the bio-recycling of mixed plastic waste with upcycling potential.

## 1. Introduction

Currently, it is estimated that nearly 80% of all plastic ever made will be discarded into landfills or as pollution within our ecosystems, and this percentage is rapidly increasing [1]. Globally, over 380 million metric tons of plastic are produced each year, and 10 million metric tons end up in our oceans annually [2]. Fifty percent of this material is made for single-use purposes, and the COVID-19 pandemic ultimately created an even greater demand for single-use plastics, which only exacerbated the plastic pollution crisis [3,4]. Continuing at this pace, it is estimated that by 2050, by weight, there will be more plastics in our oceans than fish [5]. The widespread pollution caused by plastics and microplastics is ubiquitous [1]. They have been found within humans [6] and many other organisms [7], in virtually all the environmental locations that have been tested [8,9,10], and even dispersed in rainwater [11].

Polyethylene terephthalate (PET), polyurethane (PUR), polyethylene (PE), polyamide (PA), polystyrene (PS), polyvinylchloride (PVC), and polypropylene (PP) comprise the majority of manufactured plastics [12]. Fortunately, within the microbial world, enzymes exist to degrade naturally occurring compounds, such as lignin, that are chemically similar to synthetic polymers [13,14], and the microbes encoding these enzymes are found in both marine and terrestrial environments [15]. Of these polymers, there has been a concerted effort to identify PETases to degrade recycled water bottles and other single-use containers, improve their activity through the modification of active sites, and improve thermal stability. In 2016, from a plastics recycling facility in Japan, Yoshida et al. isolated *Ideonella sakaiensis* that encoded a PETase and a second enzyme, MHETase, involved in the stepwise degradation of PET [16]. *I. sakaiensis* was determined to exist as part of a consortium of microbes and was able to use PET as a sole source of carbon and energy [17]. Taking a metagenomics approach, researchers who were studying leaf compost identified a cutinase (LCC) that was also able to degrade PET [18].

Further research shows the benefits of using microbial consortia for bioremediation due to their mixed metabolism within native bacterial communities [19]. Consortia of environmental microbiomes have been documented to enzymatically and synergistically degrade petroleum and petroleum-derived products such as plastics [20,21,22,23,24,25,26]. Previously, we isolated a microbial consortium from petroleum-contaminated soils comprised of five *Pseudomonas* and *Bacillus* spp. that showed the synergistic degradation of PET [25]. Sequencing revealed that these strains had unique and diverse genomes with many enzymes with hydrolytic activities [27]. However, the genetics behind the mixed metabolism and observed synergy amongst these strains had not been fully realized. By employing pangenomic analysis, our goal was to identify the core gene clusters responsible for PET degradation, as well as other plastic-degrading genes and pathways from our previously described consortia of bacteria [25].

A pangenome is the collective set of genes shared by all strains of a species [28]. It contains a core genome with all the genes shared by the same species, such as essential metabolic genes and regulatory functions, and a variable genome. The variable genome within a pangenome are those genes that are present in two or more strains; they are often genes from specific adaptations within their environment and are not found within the core genome [29,30]. By using reference genomes of closely related species that were not identified to degrade PET, the gene clusters shared amongst the consortia members were analyzed to give insight into their adaptive evolution toward plastic degradation [31,32].

Additionally, studying transcriptomics of consortia can be a powerful tool to identify the active RNA transcripts under restrictive conditions, such as growth on a single carbon source like PET [33,34,35]. Our results identify the genetic pathways within the *Pseudomonas* and *Bacillus* spp. that, in part, explain the synergistic degradation of PET. These data regarding bacteria residing in petroleum-polluted soils revealed a wealth of enzymes that are predicted [36] to be associated with the biodegradation of multiple types of biological and synthetic plastic polymers.

## 2. Results

### 2.1. Employing Pangenomic Analysis toward Understanding Synergistic PET Degradation

Previously, our group provided experimental evidence that a full consortium of three *Pseudomonas* and two *Bacillus* spp. collected from hydrocarbon-polluted soil could synergistically degrade polyethylene terephthalate (PET) [25]. Due to this observed synergy, we sought to examine the pangenome encoded by the full consortium to gain deeper insight into how they might be degrading PET genetically and metabolically.

In this analysis, the 232 core genomes (i.e., all *Pseudomonas* (144) and *Bacillus* (88) species within the MicroScope platform database [37] (excluding the five strains within our full consortium) at the time of this analysis) were purposefully excluded (Appendix A). The parameters were set using 50% amino acid (aa) identity and 80% alignment coverage. After the exclusion of the core genome found in all *Pseudomonas* and *Bacillus* spp. within the database, 259 gene groups were found to be shared in the core genome within the pangenome of these five strains (Figure 1).

*Bacillus albus* PFN01 strain 13.1 and *Pseudomonas* sp. B10 strain 9.2 had the most diverse set of accessory genes (Appendix A) compared to all other strains within the full consortium indicating clear differences between the consortia members. *Bacillus* strains 9.1 and 13.1 share a large pangenome of over 3305 genes. The pangenome of all three *Pseudomonas* strains contained over 2292 shared genes. When comparing *Pseudomonas* strain 10 and 13.2, it was established that strain 10 contained 70 unique, species-specific hypothetical proteins different from the pangenome, while 13.2 contained 58 different genes, despite species relation.

Initially, within the core pangenome, several gene groups were of interest, including aldehyde dehydrogenases, esterases, and alcohol dehydrogenases (Appendix A). Prior to RNA sequencing and database analysis, these genes were predicted to be involved in PET degradation, as dehydrogenases and esterases have been implicated in PET oligomer and monomer degradation, but the specific enzymes responsible had yet to be confirmed.

### 2.2. Esterase EstB in Strain 9.2 Is Active against PET

An esterase encoded in the *estB* gene identified within the pangenome was discovered and explored. Previously, researchers discovered that an *estB* encoded in *Enterobacter* sp. HY1 could degrade BHET [39]. With this in mind, a deletion mutant of esterase gene *estB* (Δ*estB*) in strain *Pseudomonas* 9.2 was created using *pEX18Tc*
*sacB* suicide plasmid to test if it could cleave BHET [40]. This Δ*estB* mutant showed significantly decreased esterase activity on 4-nitrophenyl butyrate (p-np-butyrate) (Figure 2a). To confirm p-np-butyrate activity, the purified EstB protein resulted in increased p-np-butyrate cleavage, illustrated in Figure 2b.

The purified EstB was then incubated on either BHET (Figure 3a) or micro-PET (Figure 3b). The hydrolysis byproducts from both compounds were confirmed via HPLC, with the increased absorbance at 245 nm indicating the presence of oligomer MHET and monomer TPA.

The incubation of EstB on BHET results in a statistically significant decrease in BHET peak area (*p* < 0.0001), coupled with a statistically significant increase in the MHET peak area (*p* < 0.0001) (Figure 3b). There was also a statistically significant increase in the TPA peak area (*p* < 0.0001), although this is likely a result of the spontaneous hydrolysis of MHET. This is because EstB largely shows no activity against MHET, and the hydrolytic cleavage of BHET would only yield MHET and EG, not TPA. All three monomers are present in the micro-PET solution prior to the addition of EstB, which is likely a result of hydrolysis from the micro-PET production procedure in which TFA was used to dissolve the PET plastic.

The addition of EstB to the micro-PET solution results in a statistically significant increase in both the MHET and the TPA peak areas (*p* < 0.0001), but no change in the BHET peak area. The absence of an increase in the BHET peak area indicates that any BHET freed from PET cleavage is subsequently hydrolyzed to MHET. TPA is expected to be released from the hydrolysis of PET due to different oligomer and monomer products being released, depending on which ester bonds are cleaved (Appendix A). While EG is not detected via HPLC spectra at 254 nm, it is assumed to be in the solution, as the cleavage of BHET to MHET releases EG.

In order to conduct preliminary structure analyses of EstB, its 3D structure was predicted with a computational AI system called AlphaFold v2.0 [41]. AlphaFold predicts the 3D structure of a protein from just its amino acid sequence by incorporating novel neural network architectures and training procedures based on homology to solve structural, evolutionary, physical, and geometric constraints [41]. The structure of EstB belongs to the α/β hydrolase superfamily and contains the conserved serine hydrolase Gly-X-Ser-X-Gly (Gly112- Phe113-Ser114-Gln115-Gly116) located at the active site [42], with the residues Asp168 and His199 completing the catalytic triad.

As EstB shares the same activity on PET and BHET as the PETase of *Ideonella sakaiensis* [43], degrading PET and BHET to form MHET, the two enzymes were subsequently compared on a structural level. The *IsPETase* has an L-shaped hydrophobic binding cleft that binds the PET polymer, with hydrolytic cleavage occurring at the catalytic serine residue. The dimensions of this cleft are roughly 40 Å in length and 29 Å in width. A similar hydrophobic cleft was identified in EstB, extending from the catalytic triad. The dimensions of this cleft are approximately 33 Å in length and 30 Å in width (Figure 4). Based on the mutational analysis, the assays using a purified protein, and the initial structural analysis, we concluded that EstB of *Pseudomonas* strain 9.2 is indeed a PETase.

### 2.3. Identification of Predicted Plastic Degrading Genes Using PMBD

The annotated amino acid FASTA file generated above from pangenomic analysis was used to mine for genes with plastic-degrading potential using the Plastic Metabolic Biodegradation Database (PMBD) [36]. Within the pangenome, 250 genes were predicted to be involved with plastic degradation (Appendix A). These genes consisted mostly of hydrolases and oxidoreductases of various enzyme classes (Appendix A), but also included regulatory proteins, essential transferases, lyases, and isomerases/translocases crucial to downstream monomer assimilation.

### 2.4. Genes Implicated in PET Degradation

The oligomers BHET/MHET and the polymer PET have been previously identified to be hydrolyzed via esterase/hydrolase activity. Two predicted PET hydrolases encoded in *Streptomyces* sp. shared more than 23% identity to alpha/beta hydrolases/dienelactone hydrolase encoded in *Bacillus* strain 9.1 and in *Pseudomonas* strains 10/13.2. Additionally, three enzymes within four of the five strains, the exception being strain *Bacillus* 13.1, were predicted to have feruloyl esterase activity; these enzymes were annotated as cellulose-binding proteins, with two being feruloyl esterases, according to UniProt (Table 1). The esterase EstB was one of three enzymes identified as having feruloyl esterase activity.

Monomers of PET, terephthalic acid (TPA), and ethylene glycol (EG) were previously identified as being assimilated by the full consortium [25]. Analysis shows (Table 1) that the *Pseudomonas* spp. strains encoded dioxygenases/reductase complexes with greater than 34% identity to previously identified TPA dioxygenases in *Comamonas* sp. [33], as well as other dioxygenases complexes (Table 3) that may be responsible for TPA degradation. We predicted that aldehyde dehydrogenases and/or alcohol dehydrogenases encoded within the genomes of the full consortium would be involved in the biodegradation EG. Aldehyde dehydrogenases have been implicated in poly(ethylene) glycol (PEG) and EG metabolism, as well as implicated in the surface modification of PET plastic [35,44,45] Identified from this analysis, an encoded aldehyde dehydrogenase had nearly 34% identity to petroleum-degrading *Mycobacterium vanbaalenii*. As these organisms were collected from oil-contaminated soils, we hypothesized that this or other aldehyde dehydrogenases may be active against petroleum-derived carbon sources, such as PET.

### 2.5. Genes Implicated in the Biodegradation of Other Plastic Types

Despite investigating the pangenome for PET degradation initially, a multitude of identified genes were implicated in the degradation of other plastic types. These types included synthetic PUR and PVA, as well as more biodegradable polymers and biopolymers such as PLA and PHA/PHB (Table 1). Research has shown that several *Pseudomonas* and *Bacillus* species are highly proficient at degrading most polymers and xenobiotics [46,47,48,49]. It is possible that several other plastic types might be degradable by our full consortium. Other encoded oxidoreductases and hydrolases (Appendix A) may degrade other plastic types as well, and our group currently has studies underway to determine the full consortium’s biodegradation potential.

### 2.6. Genes Implicated in the Biodegradation of Plasticizer and Xenobiotics

We observed a majority of biodegradation enzymes predominately being involved in plasticizer degradation (Table 2). This is in accordance with previous research showing that within the global microbiome, soil microbes, such as these strains, have a higher diversity of plasticizer degradation when compared to marine environments [15]. Several of these oxidoreductases and dehydrogenases were correlated with plasticizers such as polyethylene glycol (PEG) [44] and polypropylene glycol (PPG) [45], as well as xenobiotics such as PAHs and other phenols (Table 2) [50,51].

Phthalate- and paraben-degrading oxidoreductases comprised the majority of plastic-related genes (Table 3). The enzyme class numbers of the specific oxidoreductases and other enzyme types are included in Appendix A. Many genes were predicted to be involved in the degradation of both phenolic acids/phthalates [52] such as TPA [33] and parabens such as 4-hydroxybenzoic acid [53].

### 2.7. Strains within the Full Consortium Have Diverse Carbon Sources Utilization

A microbial community-based analysis was performed using Biolog EcoPlates^®^ to assess the overall carbon utilization amongst the strains and the full consortium. Results indicate a wide substrate capability (Figure 5) of the individual strains, which is also evidenced by their pangenomes (Figure 1).

The non-aromatic amino acid L-asparagine was the preferred carbon source of most strains, and as a result, this carbon source was chosen as a control for further transcriptomic analysis of PET degradation. Of the most interest, as examined above, was the paraben 4-hydroxybenzoic acid, which was within the top ten fastest utilized carbon sources for the full consortium. This carbon source is proven to be transported into the cell using PcaK [54,55] and is hypothesized to be a probable transporter for TPA in the *Pseudomonas* species.

As expected, based on previous screenings [25] for lipase and esterase activity, Tween 40 was a favorable carbon source amongst the strains. Overall, amino acids and other nitrogen-containing compounds tended to be a favorable carbon source for all strains, excluding *Bacillus* 9.1. Strain 9.1 was -growing under room temperature conditions and preferentially grew on polysaccharides and monosaccharides, as well as the chitin monomer, N-acetyl-glucosamine, and non-ionic surfactant Tween 80. This slow growth may be temperature-dependent, as *Bacillus* 9.1 grows rapidly at 40 °C. Putrescine was also preferred by the full consortium, which is of interest, as putrescine is an alkane. This provides further evidence there that the full consortium may possess alkane metabolism potentially involved in LDPE degradation [56,57].

In contrast to the observed synergy on PET, the full consortium did not degrade as wide of a range of substrates as the individual strains (with the exception of *Bacillus* strain 9.1, 77%) with a functional diversity score of 84% (see methods for diversity and similarity scoring); however, the full consortium did utilize preferred carbon sources more efficiently than all other strains, reaching an average optical density greater than the individual strains in the same time period (Appendix A) despite the same inoculation concentration and conditions. *Bacillus* strain 13.1 and the full consortium were the most similar in carbon source utilization, with 90% similarity. *Pseudomonas* strain 10 had the most diverse carbon utilization of all, with a 97% functional diversity, able to utilize all but 2-hydroxy benzoic acid. Despite *Pseudomonas* 10 and 13.2 being the same species, they were only 90% similar in carbon source utilization. This stark contrast in carbon source utilization and the differences within the pangenome (Figure 1) show incredible functional genetic diversity.

These differences might explain the observed variations in growth and biofilm formation (Figure 6). Using a crystal violet microtiter assay, differences in biofilm formation between individual strains and the full consortium were quantified. *Pseudomonas* strains 9.2 and 10 exhibited significantly more biofilm formation than strains 9.1, 13.1, 13.2, and the full consortium. Of greatest interest was the difference in biofilm production between *Pseudomonas* strains 10 and 13.2, which are the same species. After 48 h of growth at 37 °C, strain 10 exhibited a solubilized crystal violet optical density of 0.12 ± 0.035, while the optical density of strain 13.2 was only 0.03 ± 0.015. This significant difference could perhaps be attributed to the two strains’ variation in carbon source utilization, their differences in transcriptional activity, and their distinct species-specific pangenome encode proteins.

### 2.8. Differential RNAseq Analysis Shows Diverse Transcriptional Activity Amongst the Individual Strains Related to PET Monomer Degradation and PHA Storage

RNA sequencing (RNAseq) analysis provided insight into how the full consortium may be interacting and metabolizing PET and its monomers during the late exponential phase of growth. It appears the strains within the full consortium may be degrading PET in different ways, possibly explaining synergistic effects. Differential analysis obtained by using data from the full consortium grown on L-asparagine as a control compared to PET indicates a significant upregulation of genes related to initial PET degradation; the predicted monomer metabolism of both TPA and EG, as well as downstream PHA metabolism, were also observed to be upregulated amongst the strains (Table 4).

A previous transcriptomic analysis from Kumari et al. [35]. proposed that aldehyde dehydrogenases are active against PET in *Bacillus*, generating 4-[(2-hydroxyethoxy)carbonyl]benzoate resulting from the deprotonation of the free carboxy group of MHET. In this study, RNAseq analysis of the full consortium grown on PET has shown a significant upregulation of an aldehyde dehydrogenase in strain 9.2. This aldehyde dehydrogenase might be acting on EG but may also act on PET itself. As EstB was also experimentally shown to hydrolyze PET to MHET, this further supports the work of Kumari et al., who proposed that aldehyde dehydrogenases and carboxylesterases play a role in PET hydrolysis in some species [35].

Despite TPA degradation-related genes being absent in strain 9.1 [25] during previous genome exploration, based on differential analysis and annotation, strain 9.1 had significant upregulation of genes with significant identity to phthalate dioxygenases and dehydrogenase (Table 4), as well as a phenol hydrolase. A transporter with significant identity to a gentisate transporter (GenK) was also upregulated and is likely how TPA or oligomers of PET may transverse the outer membrane of strain 9.1. Additionally, in 9.1, an alcohol dehydrogenase and a glyoxal reductase were upregulated, likely illustrating its role in ethylene glycol metabolism.

A differential analysis of RNAseq transcripts from strain 10 and 13.2 suggests TPA was actively being degraded at the time of RNA extraction. Numerous dioxygenases and decarboxylases with significant identity to phthalate oxidoreductases, including Terephthalate 1,2-dioxygenase, were present (Table 4). Additionally, evidence of downstream beta-ketoadipate metabolism indicates that TPA is being further metabolized, likely toward PHA/PHB synthesis.

The upregulation of related genes and PHA biosynthesis were present in all strain transcripts (Table 4). Both *Pseudomonas* strains 10 and 13.2 had significant transcriptional upregulation of carboxylesterase NlhH, which was identified within the pangenome above as likely a PHA/PHB depolymerase [58]. The deletion of *nlhH* in strain 9.2 showed no reduction of np-butyrate hydrolysis (Appendix A), indicating differences in esterase activity compared to EstB and may or may not be directly involved in PET polymer depolymerization.

Interestingly, only strain 10 had an increased transcription level of surfactin (Table 4), and 113 ‘hypothetical proteins’ were upregulated in strain 10, whereas only 77 were upregulated in strain 13.2 (Appendix A). These gene differences in strain 10 may partially explain the differences in biofilm production [59] (Figure 6).

## 3. Discussion

By using the pangenome of the full consortium containing three *Pseudomonas* and two *Bacillus* strains, we were able to predict which enzymes were potential PET degradation enzymes. From this analysis, many hydrolases, oxidoreductases, and dehydrogenases were identified (Appendix A). Previously, a group determined that a secreted carboxylesterase 2, EstB encoded in *Enterobacter* sp. HY1, was able to degrade BHET, a monomer of PET [39]. As our group identified an EstB within the pangenome, we hypothesized this enzyme within the *Pseudomonas* strains might be a PET/BHET hydrolyzing enzyme. After screening for p-np-butyrate esterase activity via the deletion and purification of EstB (Figure 2), as well as incubation on BHET and PET, our data indicated that EstB hydrolyzed BHET to the oligomer MHET and the monomer TPA (Figure 3), similar to the *Ideonella sakaiensis* PETase [16].

A comparison of the structure of *IsPETase* [60] to the predicted structure of EstB revealed considerable structural similarity, including a similar binding cleft, catalytic triad, and lack of a lid structure (Figure 4), indicating that EstB is a PETase. The observations of similar active sites and enzymatic activities, combined with a relatively low primary amino acid sequence identity between the *IsPETase* and EstB, is consistent with the idea of the convergent evolution of bacteria in disparate locations to evolve the ability to degrade PET and other plastics. EstB was predicted to have feruloyl esterase activity when aligned with the PMBD (Table 1), and other enzymes that could potentially degrade PET were also identified.

The predicted PETases included two dienelactone hydrolases encoded in both *Pseudomonas* strains 10 and 13.2 and in *Bacillus* strain 9.1. Further transcriptomic analysis within the *Pseudomonas* strains indicated the upregulation of carboxylesterases NlhH in strains 10 and 13.2 and an aldehyde dehydrogenase upregulated in 9.2 (Table 4) when the strains were incubated on PET. Kumari et al. previously identified, through transcriptomic analysis, the complex interactions of carboxylesterases and aldehyde dehydrogenases on the degradation of PET, possibly through the generation of 4-[(2-hydroxyethoxy) carbonyl] [35]. This study gives further evidence of potential aldehyde dehydrogenase- and carboxylesterase-mediated PET degradation.

While the *Pseudomonas* strains seemed to be actively metabolizing PET and TPA using the presence of various oxidoreductases and dehydrogenases (Table 4), *Bacillus* strain 9.1 seems to play a key role in ethylene glycol metabolism. Strain 9.1 had a high upregulation of an alcohol dehydrogenase and glyoxalase, which are two genes implicated in EG metabolism. However, strain 9.1 also might be involved in BHET and TPA metabolism. The upregulation of the aromatic transporters GenK in *Bacillus* strain 9.1 may be responsible for PET monomer uptake, such as TPA. The downstream metabolism of TPA was observed within the *Pseudomonas* species via the presence of genes involving Beta-ketoadipate, butonate, and polyhydroxyalkanoate (PHA) metabolism.

It is likely the consortia utilize stored biopolymers PHA/PHB as a supplementary carbon source over time. After hydrolyzing the PET polymer and oligomers, PET monomers are assimilated, and the consortia generate biopolymer PHA. Evidence of PHA synthesis and depolymerization is shown in the upregulation of PHA/PHB depolymerases and synthases (Table 4) when grown on PET. It is hypothesized that there is an inability to cleave PET further after reaching the late exponential/stationary phase, either through the buildup of toxic intermediaries or product inhibition. This storage of PHA/PHB provides potential “upcycling” of PET waste as an alternative biodegradable plastic.

One important aspect of PET degradation is the ability of the bacteria to adhere to the surface of the hydrophobic polymer. Biofilms are crucial for PET degradation due to increased adherence to the surface of PET, accompanied by increased hydrophilicity and carbonyl index [61]. Biofilm production was measured in all consortia members (Figure 6), and the data indicated that *Pseudomonas* strain 10 possessed the greatest biofilm potential, compared to the other strains. Surfactin production directly results in increased biofilm formation [59], and we observed the presence of and significant upregulation of surfactin in strain 10, which is consistent with the increased biofilm formation capabilities. This was not observed in *Pseudomonas* strain 13.2 or the other consortium members.

The transcripts between all strains, particularly *Pseudomonas* 10 and 13.2, that were grown on plastic varied dramatically. Interestingly, strain 10 contained 70 unique species-specific hypothetical proteins different from the pangenome, while 13.2 contained 58, even though they are genetically the same species. This might, in part, explain the observed differences in growth and gene expression (Table 4) and biofilm formation (Figure 6). These data provide insight into how the consortia might be interacting when grown on the surface of PET, showing different functional roles. Of note, *Bacillus* strain 13.1 had very few transcripts, possibly indicating a temporal function in early phases of PET biodegradation but not in later stages.

While able to degrade PET, this environmental consortium is also highly carbon diverse; the strains are able to degrade a wide variety of carbon sources, illustrated by the Biolog EcoPlate data (Figure 5) and the large number of predicted polymer biodegradation genes (Table 1 and Table 2). The full consortium appears to have a pangenome capable of degrading a variety of plastic types, which would make mixed plastic recycling possible. Polyurethane [24], LDPE [62,63] and PLA [64,65] degradation have been observed by both *Pseudomonas* and *Bacillus* spp. previously. PVA, a water-soluble plastic that is typically used in dish and laundry detergent “pods” and has been entering waterways and ecosystems due to unprecedented commercial use [66,67], was also predicted to be degradable by the full consortium.

Importantly, not only do these strains contain predicted genes capable of degrading multiple plastic types, but they are also predicted to degrade common plasticizers, including phthalates, paraben, and other aromatic/phenolic compounds. These plasticizers leach and are ecotoxic, unlike most inert plastics [68]. This illustrates the capability of consortia to degrade mixed plastic types, including those with certain additives. Plasticizers are crucial in the discussion of plastic wastes, as they leach over time [69,70] and are of concern, as research has shown them to be acutely toxic to humans and the environment [71,72,73]. Additives have historically been largely unregulated [74], and many are not required to be disclosed or labeled. While certain additives have shown to be biodegradable [75], many are not, as many products seek to be anti-microbial.

These plasticizers pose challenges for not only biological recycling efforts but also for conventional chemical and mechanical recycling, as they limit how many times a plastic can be recycled, if at all [74]. Even though microorganisms are slowly and globally utilizing plastics as a carbon source, without a collaborative effort, the plastic pollution problem will only worsen. Luckily, some efforts to increase the regulation of and research into plasticizers are ongoing [76]. Research surrounding plasticizers’ environmental toxicity and biodegradability, if intended to be part of single-use products, should be fully considered. With the proper governmental regulation of non-toxic plastic additives amenable to biodegradation [75], it is possible to pursue biological consortia-based approaches toward plastic recycling and bio-upcycling efforts.

## 4. Materials and Methods

### 4.1. Omics Approaches

#### 4.1.1. Pangenome Analysis

Genomes deposited in GenBank (BioProject Accession: PRJNA517285) were uploaded to the MicroScope (LABGeM, Courcouronnes, FRA) [37] website for expert annotation and comparative genomic analysis using the Pan-genome Analysis tool. Pangenomes and core/variable genomes were generated using MicroScope gene families (MICFAM) computed using SiLiX software [38]. MICFAM parameters were set to 50% aa identity/80% alignment coverage. The core genomes of 232 genomes (i.e., all *Pseudomonas* (144) and *Bacillus* (88) species, within the MicroScope platform database (excluding the five strains within the full consortium) at the time of this analysis) were excluded from the analysis. Appendix A contains a phylogram of all species included in the analysis.

#### 4.1.2. PMBD Analysis

The PMBD database is comprised of 949 microorganisms with 79 genes indicated in the biodegradation of plastics and more than 8000 annotated enzymes/proteins predicted to be involved in plastics biodegradation [36]. The annotated pangenomes of all five strains within the full consortium were obtained from the MicroScope platform [37]. These sequences were BLASTED against the PMBD database; e-value cutoff >0.001 and sequences with greater than 20% percent identity were considered significant alignments.

### 4.2. RNA Sequencing to Determine Genes Implicated in PET Biodegradation

#### 4.2.1. Total RNA Extraction and Sequencing

Bacterial cultures of the full consortium were prepared in Liquid Carbon Free Media (LCFBM) as published [25] with either 1% (*w/v*) granular PET or L-asparagine. Optical Density measurements at 600 nm (OD_600_) were used to determine when the culture had reached mid to late exponential phase of growth. L-asparagine was harvested after reaching exponential phase at an OD600 of 0.300 A, and the PET samples were harvested once they reached an optical density of 0.200 A. Cells were pelleted and resuspended in RNAprotect bacteria reagent (Qiagen, Germantown, MD, USA) to prevent RNA degradation and frozen at –80 °C until time of extraction.

As extraction of total RNA from mixed culture of bacteria may present bias against species within the culture, RNA extraction was conducted according to the GTC method and DNA removal previously described by Stark et al. 2014 for the most efficient extraction rate for both *Bacillus* and *Pseudomonas* [77]. All chemicals required were purchased from Sigma-Aldrich (Sigma-Aldrich, St. Louis, MO, USA). Total RNA extractions were tested for purity and quantity using a Nanodrop™ 1000 Spectrophotometer (Thermo Fisher Waltham, MA, USA) and Qubit™ 4 Fluorometer (Invitrogen, Waltham, MA, USA). Illumina prokaryotic RNA sequencing was completed by Novogene Beijing Bioinformatics Technology, Co., Ltd. Illumina NovaSeq 6000 (Novogene, Beijing, CHN).

#### 4.2.2. RNAseq Analysis

Paired-end Illumina RNA sequences were generated for two control and two experimental samples. The quality of the raw reads was assessed with FastQC v0.11.3 [78]. After QC v0.9.6 was run to separate and retain high-quality sequences [79], high-quality transcripts were then used to recruit onto each individual genome within the consortium using Bowtie2 v2.3.5.1 [80]. SAM files were generated using Samtools v1.10 [81]. Expression was calculated and normalized using FeatureCounts v2.0.1 [82]. Once expression was calculated, differential expression between the control and the experimental treatments was calculated using the Bioconductor/R packages DESeq2 v1.34.0 and EdgeR v3.36.9 [83,84]. Upregulated genes for each individual genome were analyzed at a cut-off e-value of 0.05. Specific genes/enzymes associated with PET degradation were selected if they were upregulated in relation to the control.

### 4.3. Evaluation of Carbon Source Utilization and Strain Biofilm Production

#### 4.3.1. Biolog EcoPlates

The growth of overnight bacterial cultures grown in LB was measured via OD_600_. They were rinsed as previously described [25] and diluted to 0.1 A before their addition to each plate. Observance of purple color was indicative of respiration, as the cells reduce the tetrazolium dye included within each carbon source in the plate. Thirty-one carbon sources, in triplicate, were evaluated kinetically according to the manufacturer’s instruction (Biolog EcoPlate, Hayward, CA, USA) using a colorimetric assay (clear to purple, OD_590_) via a Tecan infinite M200PRO microplate reader (Tecan, Zürich, CHE) over a 24-h period (over 80,000 s) at room temperature (25 ± 2 °C). Triplicate values were averaged in Excel (Microsoft Corporation, Redmond, WA, USA) analysis of relative absorbance for each sample over time, which was determined via subtraction of the control (water) to account for spontaneous tetrazolium dye formation.

To examine quantitative relationships between the strains and their ability to utilize the 31 unique carbon sources, two calculations were performed to determine the functional diversity based on all 31 carbon sources and the strain’s/consortium’s relative similarity (Ssm).

% Functional Diversity = (a ÷ 31) × 100.

% Similarity (Ssm) = ((a + d) ÷ (a + b + c + d)) × 100.

a = the number of carbon sources used by strains and consortia.

b = the number of carbon sources used the strain of interest only.

c = the number of carbon sources used by the full consortium/other strain only.

d = the number of carbon sources not utilized by either.

#### 4.3.2. Microtiter Plate Biofilm Quantification Assay

Growth of overnight bacterial cultures for individual strains incubated in LB at 26 °C was measured via OD_600_. Cells were rinsed, followed by dilution in M63 media to OD_600_ = 1. A suspension of the full consortium was created by combining equal volumes of the five individual strain suspensions. These suspensions were further diluted to 1:100 in M63 supplemented with 0.2% ethanol, 1 mM MgSO_4_, and 0.5% casamino acids and transferred to a polystyrene FALCON non-tissue culture treated microtiter plate (Corning, Corning, New York, USA) (100 µL/well). Negative controls consisted of supplemented M63 only. After 48 h of incubation at 26 °C, planktonic growth was removed, and the wells were gently washed with 125 µL of sterile phosphate-buffered saline (PBS). Biofilms were stained with 0.1% crystal violet (125 µL/well) for 15 min and then washed three times with 125 µL PBS. The remaining dye was solubilized with 100% ethanol (200 µL/well) for 15 min, then 100 µL/well was transferred to a new polystyrene microtiter plate. The optical density of solution in each well was measured at 600 nm using a microplate reader (Infinite 200, TECAN, Grödig, Salzburg, AUT).

### 4.4. Mutant Preparation of ΔestB

Single gene deletion of *estB* in Pseudomonas strain 9.2 was performed as previously described [40]. Briefly, primers were designed to amplify fragments upstream and downstream of *estB* (upstream forward, 5′-ccgggtaccgagctcgaattCACGATCGCCAGAGTGGATT-3′ and upstream reverse, 5′-ggcgacccctgtcgcaattgGGCTGCTCCAATTGTGTGCG-3′; downstream forward, 5′-cgcacacaattggagcagccCAATTGCGACAGGGGTCGCC-3′ and downstream reverse, 5′-tatgaccatgattacgaattTAACCCAGCACCTGAGCCTG-3′, uppercase indicates annealing segment). These primers contain overhang segments to facilitate ligation into the vector *pEX18Tc* using an NEB Gibson Assembly kit (New England Biolabs, Ipswich, MA, USA). Correct assembly of the vector was confirmed by amplifying the MCS using *pEX18Tc* universal primers (forward, 5′-GGCTCGTATGTTGTGTGGAATTGTG-3′ and reverse, 5′-GGATGTGCTGCAAGGCGATTAAG-3′) and sequencing that fragment (ACGT Sequencing, Wheeling, IL, USA). Assembled vector with flanking *estB* fragments (pRDEXestB) was transformed into *Pseudomonas* strain 9.2, and merodiploids were selected by growth on LB-tetracycline (15 µg/mL). Deletion of *estB* was induced by growing merodiploids of sucrose plates (TYS10) and was confirmed by amplifying and sequencing a fragment spanning slightly upstream and downstream of *estB* (forward, 5′-CGCTGACACCCAGTACTGCAGC-3′ and reverse, 5′-GATCGAGATCAGGAACGCGGCG-3′). All PCR conducted utilized a touchdown protocol with Q5 polymerase.

### 4.5. EstB Purification

#### 4.5.1. Amplification and Purification of estB Vector Insert

Primers were designed using Benchling (Benchling, San Francisco, CA, USA) to amplify estB without its start or stop codon in order to express EstB with a C-terminal T7 tag and an N-terminus 6xHis tag. These designed primers included partial overhangs to allow for assembly into the expression vector. The primers are as follows: forward, 5′-atgggtcgcggatccgaattGACCGAGCCCTTGATTCTTCAGCCC-3′, reverse, 5′-ttgtcgacggagctcgaattGCGCAGGCGTTCGCTCAACCAT-3′ (uppercase characters indicates annealing segment). PCR was performed using genomic DNA obtained by incubating one colony of Pseudomonas strain 9.2 in 100 μL DNase-free water at 100 °C for 5 min (2 μL of this solution was used for a 20 μL PCR reaction). Each PCR contained 0.5 μM of forward and reverse primers, 200 μM dNTPs, 1× Q5 high GC enhancer, and 0.02 U/μL Q5 DNA polymerase in a 1× Q5 reaction buffer (New England Biolabs, Ipswich, MA). The thermocycler was programmed with a touchdown PCR protocol as follows: 98 °C for 1 min, 16 cycles of 98 °C for 10 s, 68 °C for 30 s with a 0.5 °C decrease every cycle, 72 °C for 20 s, followed by 30 cycles of 98 °C for 10 s, 60 °C for 30 s, 72 °C for 20 s, followed by a final extension of 72 °C for 2 min. Fragments were confirmed through gel electrophoresis and extracted and purified using a gel extraction kit (QIAGEN, Germantown, MD, USA).

#### 4.5.2. Assembly, Isolation, and Purification of RDpET24estB Expression Vector

The expression vector pET-24a(+) was linear by restriction digest with EcoRI. Gibson assembly was performed using reagents from NEB Gibson Assembly Cloning Kit. The reaction mix contained 1× Gibson master mix and a 4:1 molar ratio of insert to linearized plasmid (0.116 pmol estB insert and 0.029 pmol linearized pET-24a(+) vector). The reaction mixture was incubated at 50 °C for 15 min, then transformed into NEB 5-alpha *E. coli* via heat shock and plated on LB-kanamycin agar media. Transformants were inoculated in LB-kanamycin media, and plasmid DNA was extracted using a QIAGEN plasmid isolation kit. PCR was performed on both the transformant and the purified plasmid to confirm that the vector was correctly assembled. The primers used are the T7 universal primers for the pET-24a(+) vector: forward, 5′-TAATAACGACTCACTAATAGG-3′, reverse, 5′-GCTAGTTATTGCTCAGCGG-3′. PCR was performed using either genomic DNA obtained by incubating one colony of NEB 5-alpha *E. coli* in 100 μL DNase-free water at 100 °C for 5 min (2 μL of this solution was used for a 20 μL PCR reaction) or 10 ng of purified plasmid. Each PCR contained 0.5 μM of forward and reverse primers, 200 μM dNTPs, 1× Q5 high GC enhancer, 0.02 U/μL Q5 DNA polymerase in a 1× Q5 reaction buffer. The thermocycler was programmed with a touchdown PCR protocol as follows: 98 °C for 1 min, 10 cycles of 98 °C for 10 s, 60 °C for 30 s with a 1 °C decrease every cycle, 72 °C for 20 s, followed by 30 cycles of 98 °C for 10 s, 50 °C for 30 s, 72 °C for 20 s, followed by a final extension of 72 °C for 2 min. Fragments were confirmed through gel electrophoresis and purified using a QIAGEN PCR purification kit.

#### 4.5.3. Electroporation of BL21 E. Coli

An overnight culture of *BL21 E. coli* (8 mL, OD600 = 0.5) was pelleted by centrifugation (8000 rpm) for 10 min. Supernatant was discarded, and pellet was washed (3 × 1 mL ice-cold 10% glycerol) before being resuspended in 200 μL 10% glycerol. Roughly 0.5 μg of RDpET24estB vector was added to 50 μL cell suspension and incubated on ice for 1 min. Mixture was transferred to a pre-chilled electroporation cuvette and pulsed at 2.2 kV for 5.7 ms using a Bio-Rad MicroPulser (Bio-Rad. Hercules, CA, USA) before being transferred to 1 mL S.O.C. media and incubated in a 37 °C shaker for 1 h. Cells were plated on three individual LB agar plates containing 30 μg/mL kanamycin as follows: 50 and 100 μL of the solution, followed by pelleting the remaining solution via centrifugation (12,000 rpm) and reducing the total supernatant volume to 100 μL before resuspending the pellet and plating that solution. The plates were incubated at 37 °C overnight. Transformants were confirmed by PCR and gel electrophoresis (procedure identical to the PCR performed in the previous section).

#### 4.5.4. Expression and Purification of EstB

Two flasks containing 250 mL LB media with 30 μg/mL kanamycin were inoculated with BL21 E. coli containing the RDpET24estB expression vector and were grown shaking at 37 °C to an OD600 of 0.6. Once the desired OD600 was achieved, IPTG was added to each flask to a concentration of 0.1 mM, and the cultures were grown shaking at 16 °C overnight. Cells were harvested by centrifugation (5000 rpm) at 4 °C, supernatant was discarded, and pellet was resuspended in 30 mL lysis buffer (50 mM NaPO4, 300 mM NaCl, pH 8). Half a cOmplete™ protease inhibitor cocktail tablet (Roche, Mannheim, DEU) and lysozyme from chicken egg white (Sigma-Aldrich, St. Louis, MO, USA) to 0.4 mg/mL was added, and the mixture was incubated on ice for 20 min, vortexing frequently to ensure the protease tablet dissolved. Cells were lysed via sonication (6 × 10 s on, 20 s off, 70% power) before the solution was centrifuged (16,000 rpm) for 30 min at 4 °C. Roughly 10 mL of QIAGEN Ni-NTA agarose solution was pelleted by centrifugation (1600 rpm) for 5 min at 4 °C. Resin was allowed to settle for 5 min before discarding supernatant. The resin was washed (3 × 10 mL water, 1600 rpm centrifugation) before a final wash with 10 mL lysis buffer. Lysate was added to the resin and allowed to bind by gently shaking and inverting the tube for 1 h at 4 °C. Lysate-resin mixture was applied to a column and allowed to empty via gravity. Column was washed first with 5× column volumes (CVs) of wash buffer A (50 mM NaPO4, 300 mM NaCl, 10 mM imidazole, pH 8), then 3× CVs of wash buffer B (50 mM NaPO4, 300 mM NaCl, 20 mM imidazole, pH 8). Elution was done by adding a 3× CVs elution buffer (50 mM NaPO4, 300 mM NaCl, 250 mM imidazole, pH 8). The flow-through from each CV added was collected in a different test tube and stored at 4 °C. Resin was washed 2× CVs elution buffer, then 2× CVs water, before being stored in 10 mL 20% ethanol at 4 °C. Purification of EstB was confirmed through SDS PAGE. Elutions containing only protein were pooled and dialyzed into 50 mM Tris-HCl, 150 mM NaCl, pH 8.0 buffer. Protein concentration was determined by measuring the absorbance at 280 nm with a NanoDrop-1000 (Thermo Fisher Waltham, MA, USA), and through a BCA Assay Kit (Thermo Fischer Scientific, Waltham, MA, USA). Aliquots of protein solutions in 25% glycerol and 1 mM DTT were stored at −20 °C.

### 4.6. Screening Purified EstB and Mutant ΔestB for Esterase Activity

*Mutants*: Esterase activity was quantified based on the absorbance of 4-nitrophenol at 402 nm released from the hydrolysis of 4-nitrophenol butyrate [39]. The following reactions were performed in 200 μL volumes and in triplicate: just buffer (LCFBM), buffer with 4-nitrophenol butyrate (1.2 mM), and buffer with 4-nitrophenol butyrate and culture (1.2 mM, OD_600_ = 0.1 in reaction volume). The microplate reader (Tecan) was set to 10 cycles of 30 min at room temperature and measuring absorbance at 402 nm.

*Enzyme*: Esterase activity was quantified based on the absorbance of 4-nitrophenol at 402 nm released from the hydrolysis of 4-nitrophenol butyrate. In a solution of acetonitrile:isopropanol (1:4 *v/v*) 4-nitrophenol butyrate (Sigma-Aldrich, St. Louis, MO, USA) was solubilized to 20 mM and stored at 4 °C. The following reaction conditions were performed in 200 μL reaction volumes in a clear 96-well plate (Corning, Corning, NY), with each reaction performed in triplicate: just buffer (50 mM HCl, 1 mM CaCl_2_, pH 7.5), buffer with 4-nitrophenol butyrate (1 mM), buffer with *estB* (5 nM), and buffer with 4-nitrophenol butyrate (1 mM) and *estB* (5 nM). Absorbance of each well was measured every min for 30 min at 402 nm by a microplate reader (Tecan).

### 4.7. Screening for PET and BHET Degradation

#### 4.7.1. Synthesis of microPET and nanoPET

MicroPET was synthesized as outlined by Rodríguez-Hernández et al. [85] in order to increase the exposed surface area of the material and enhance degradation. Post-consumer PET soda bottles (Coke and Pepsi) were cut into 1 cm strips of various lengths, discarding the caps and taking care to discard any labels. A total of 50 g of post-consumer PET strips were placed into a sterilized stainless steel blender (Waring Commercial, Stamford, CT, USA), and liquid N_2_ was added until it fully covered the strips. The blender was set to high, and the liquid nitrogen was replaced until the PET was ground into a mixture of dust and small, jagged pieces. A total of 1 g of this ground PET mix was added to a beaker containing 10 mL of 90% (*v/v*) trifluoroacetic acid (TFA) and stirred for an hour until the PET was fully dissolved. Following the solvation of PET, 10 mL of 20% (*v/v*) TFA was added to the beaker and left overnight to let the PET crystalize out of solution. The mixture was then centrifuged at 6000 RCF (2496 G) for 1 h and the supernatant was discarded and replaced with 100 mL of a resuspension solution composed of 0.5% (*w/w*) sodium dodecyl sulfate. This mixture was ultrasonicated (5×, t = 10 s) via Misonix 3000 sonicator (Misonix, Farmingdale, NY, USA) until it became a milky white color, after which the microPET was allowed to settle to the bottom of the bottle overnight. The upper aqueous layer was carefully decanted to separate the nanoPET suspension from the microPET present in the bottom layer.

#### 4.7.2. BHET and PET Hydrolysis Assay

Degradation of BHET by EstB was assayed as previously described [86]. BHET buffer was made by adding BHET (Sigma-Aldrich, St. Louis, MO, USA) to 1 mM in 40 mM NaH2PO4, 80 mM NaCl, 20% (*v/v*) DMSO, pH 7.5 and stirred overnight to dissolve. Complete dissolution of the BHET particles was not observed, but their size was dramatically reduced to an estimated microparticle size (<300 µm). EstB (50 µg) was added to 1 mL of BHET buffer and incubated shaking at 40 °C overnight. Blank assays, in which no protein was added, were used as controls. All assays were performed in triplicate.

The concentration of SDS in the microPET suspension was dramatically reduced by centrifuging the microPET suspension at 12,000 rpm and removing the supernatant before adding an equal volume of Ultrapure water. This procedure was repeated three times to yield a microPET suspension with a very low SDS concentration. EstB (100 µg) was added to 1 mL of PET buffer (40 mM Tris-HCl, 0.8 mM CaCl2, 1 mL washed microPET, 20% (*v/v*) DMSO, pH 8) and incubated shaking at 40 °C overnight. Blank assays, in which no protein was added, were used as controls. All assays were performed in triplicate.

#### 4.7.3. PET Monomer Analysis Using HPLC

Analysis of PET oligomers and monomers were adapted from Furukawa et al. 2019 [87] using an Agilent Technologies 1100 series HPLC equipped with a ZORBAX Eclipse Plus C18 (Rapid Resolution, 4.6 × 100 mm 3.5 Micron) column (Agilent Technologies, Santa Clara, CA, USA). The flow rate, mobile phase, and oven temperature were as previously described [87]. Hydrolyzed products were observed at 254 nM, with a reference at 354 nM. TPA and BHET standards were used to detect peak retention times. The column used in this experiment resulted in significantly shorter retention times compared to previous literature values. Retention times on average for TPA at 1.5 min and BHET at 1.9 min. Experimentally, MHET was expected around 1.7 min.

## 5. Conclusions

Plastic pollution is choking our planet, with more of these materials being made and released into marine and terrestrial environments every year. Though occurring at a slow pace, microbes are able to degrade these man-made polymers. Species of bacteria, including *Pseudomonas* and *Bacillus*, have been implicated in the biodegradation of PET plastic, but very few specific enzymes within these organisms have been identified and characterized [88] (ref). Here we use “omics” approaches and existing databases to elucidate the genetic basis of how a consortium of soil bacteria can cooperate to degrade PET plastic, a model for how this may be occurring in the environment. Using molecular techniques, we identified a new EstB PETase encoded in *Pseudomonas* spp., an initial depolymerizer, with the potential for additional PETases to be identified and characterized within the pangenome. Unlike bacteria isolated from plastic in marine environments, these *Pseudomonas* and *Bacillus* bacteria isolated from petroleum polluted soils have robust plastic- and plasticizer-degrading capabilities, including 250 associated enzymes among five strains, demonstrating the potential for biodegradation of mixed plastic waste.

## Figures and Tables

**Figure 1 ijms-23-05612-f001:**
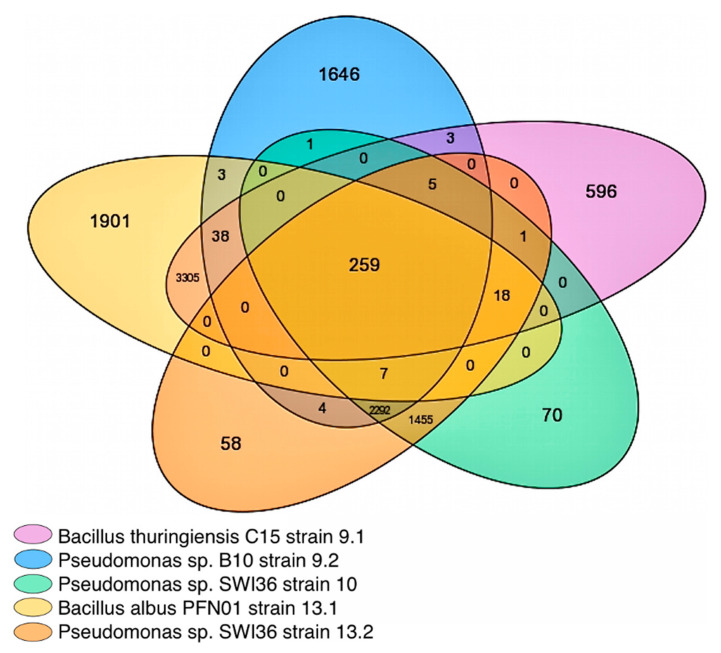
Venn diagram illustrating the number of genes within the pangenome of all five bacterial strains. Pangenome gene clusters were analyzed using MicroScope gene families (MICFAM), computed with the SiLiX softwares [38]. Genes were considered orthologs if genes contained >50% amino acid sequence similarity and 80% alignment coverage. Diagram generated by MicroScope platform with Creative Commons Attribution 4.0 International License, public use.

**Figure 2 ijms-23-05612-f002:**
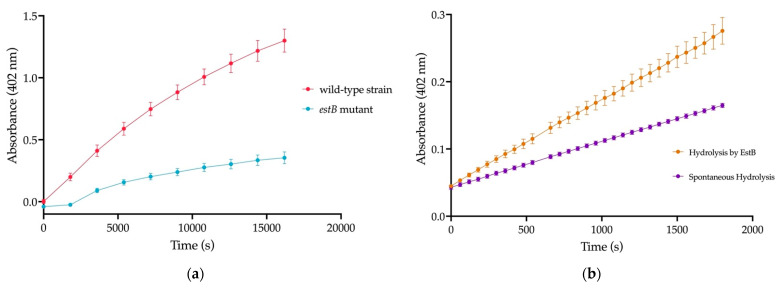
4-Nitrophenyl Butyrate (p-np butyrate) Assay to determine esterase activity (**a**) Deletion of *estB* resulted in significant decrease in activity against np-butyrate indicating loss of esterase activity; (**b**) Purified EstB resulted in significant hydrolysis of np-butyrate, indicating this protein possesses esterase activity.

**Figure 3 ijms-23-05612-f003:**
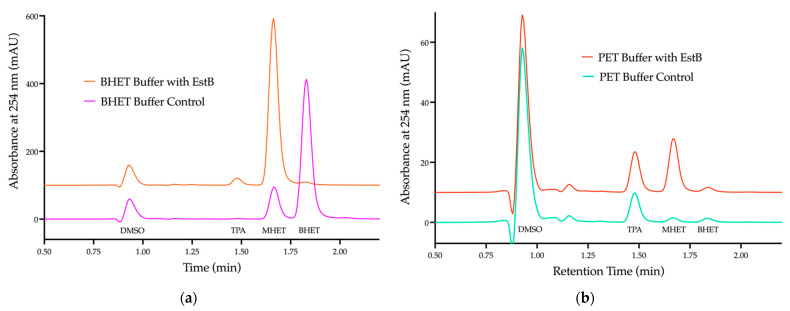
HPLC chromatogram of hydrolysis byproducts from purified EstB incubated on BHET and micro-PET (**a**) Protein EstB incubated on BHET showing significant MHET and TPA accumulation (*p* < 0.0001) (**b**) Protein EstB incubated on PET showing significant MHET accumulation (*p* < 0.0001). All samples were conducted in triplicate and averaged, there was no significant difference between replicates.

**Figure 4 ijms-23-05612-f004:**
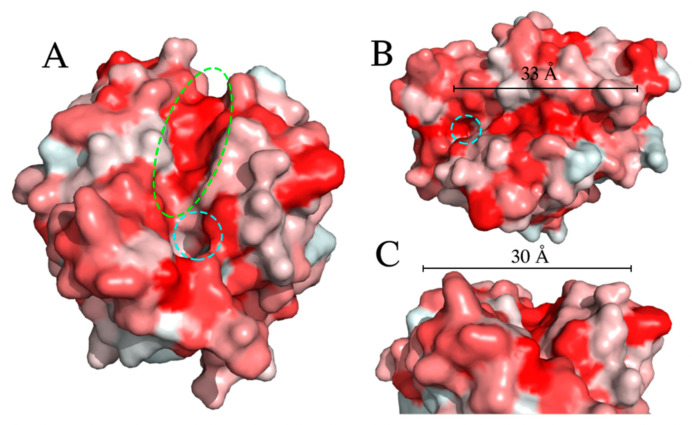
Hydrophobicity surface model of EstB. Red indicates hydrophobic regions; white indicates hydrophilic regions. (**A**) Front view of the potential substrate binding site of EstB. Catalytic residue Ser114 is indicated by dotted cyan circle, binding cleft is indicated dotted green circle. (**B**) Length of binding cleft. Catalytic residue S114 is indicated by dotted cyan circle. (**C**) Width of binding cleft.

**Figure 5 ijms-23-05612-f005:**
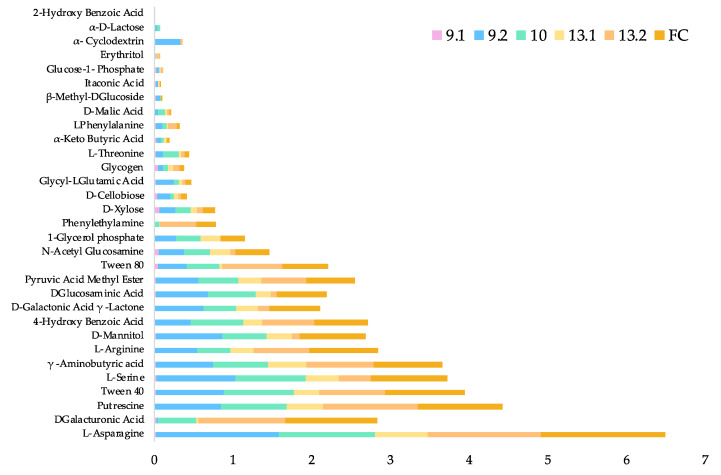
Comparison of relative carbon source utilization between the individual strains and the full consortium over 24-h using Biolog EcoPlates^®^. Thirty-one carbon sources, in triplicate, were evaluated kinetically via a colorimetric assay over a 24-h period at room temperature (25 ± 2 °C). Triplicate values were averaged to determine relative absorbance compared to the control samples (water).

**Figure 6 ijms-23-05612-f006:**
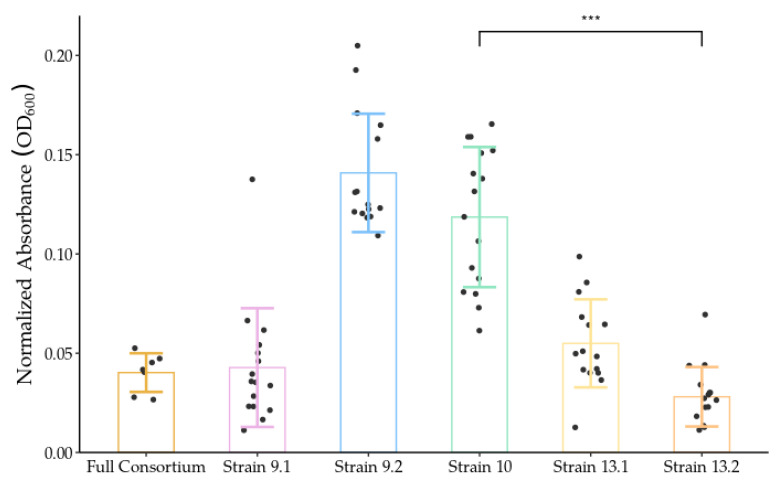
Quantification of biofilm production using crystal violet stain. Comparison of biofilm formation between the individual strains and the full consortium grown for 48 h on polystyrene 96-well plates. Biofilm production was quantified, minus background absorbance, by measuring optical density of biofilms stained with crystal violet at 600 nm using a TECAN infinite 200. *** *p* < 0.0001. Error bars denote one standard deviation from the mean.

**Table 1 ijms-23-05612-t001:** Predicted plastic biodegradation enzymes with significant percentage identity and similarity to genes encoded within the pangenome of the full consortium.

Plastic	Enzyme	Species	Uniprot ID	% Identity	E-Value	Bit-Score	Strain
**PET**	Aldehyde dehydrogenase	* Mycobacterium vanbaalenii *	Q9KHU2	33.97	8.23 × 10^−74^	239	all
	Cellulose-binding protein	* Micromonospora rifamycinica *	A0A120F7D2	27.98	8.59 × 10^−4^	37.7	10/13.2
	Feruloyl Esterase	* Phialocephala subalpina *	A0A1L7XXB0	32.00	1.43 × 10^−4^	39.7	9.2
	Feruloyl Esterase	* Rhynchosporium secalis *	A0A1E1MSN7	27.89	5.65 × 10^−5^	42.7	9.1
	Poly(ethylene terephthalate) hydrolase	* Streptomyces * sp. *111WW2*	A0A2P8AA05	23.77	2.93 × 10^−5^	43.1	10/13.2
	Poly(ethylene terephthalate) hydrolase	* Streptomyces * sp. *MH60*	A0A2S6X119	23.75	7.52 × 10^−6^	45.4	9.1
	Putative terephthalate 1,2-dioxygenase	* Rhodococcus * sp. *DK17*	Q6REK1	32.30	6.62 × 10^−33^	127	9.2/10/13.2
	TPA 1,2-dioxygenase, reductase component 1	* Comamonas * sp.	TPDR1	34.92	3.46 × 10^−5^	43.1	9.2/10/13.2
	TPA 1,2-dioxygenase, reductase component 2	* Comamonas * sp.	TPDR2	36.51	6.73 × 10^−6^	45.4	9.2/10/13.2
	TPA 1,2-dioxygenase, oxygenase alpha 1	* Comamonas * sp.	TPDA1	34.10	2.15 × 10^−29^	117	9.2/10/13.2
	TPA 1,2-dioxygenase oxygenase alpha 2	* Comamonas * sp.	TPDA2	34.10	2.21 × 10^−29^	117	9.2/10/13.2
	Twin-arginine translocation pathway signal	* Polaromonas * sp. *strain JS666*	Q12BN2	39.46	5.89 × 10^−25^	103	9.2/10/13.2
	Glyoxalase	* Azoarcus * sp. *PA01*	A0A0M0FWC0	26.471	1.19 × 10^−4^	37.7	9.1/13.1
**PLA**	PLA depolymerase	uncultured bacterium	A4UZ10	46.71	1.49 × 10^−124^	365	9.1/13.1
	PLA depolymerase	uncultured bacterium	A4UZ14	37.46	6.05 × 10^−67^	211	9.2/10/13.2
	PLA depolymerase (Fragment)	uncultured bacterium	A4UZ11	48.5	1.80 × 10^−123^	362	9.1/9.2/13.1
**PUR**	Polyurethanase	* Pseudomonas * sp. *FW305-BF6*	A0A2N8H9Y3	99.51	0.00	1225	9.2
	Polyurethanase	* Pseudomonas fluorescens *	A0A0C2A4M0	59.52	8.28 × 10^−7^	52	10/13.2
	Polyurethanase (Fragment)	* Pseudomonas * sp. *DrBHI1*	A0A246GXG4	66.61	0.00	733	9.2/10/13.2
	Polyurethanase A	* Pseudomonas * sp. *Os17*	A0A0D6BHI0	77.96	0.00	994	9.2
**PVA**	Polyvinyl alcohol dehydrogenase	* Pseudomonas * sp. *FW305-BF6*	A0A2N8GY02	98.23	0.00	561	9.2
	Polyvinyl alcohol dehydrogenase	* Xanthomonas arboricola *	A0A2S6Y8G0	37.5	4.65 × 10^−4^	41.2	10/13.2
	Polyvinyl alcohol dehydrogenase (Fragment)	* Opitutae bacterium *	A0A2D6VIL8	27.63	1.89 × 10^−4^	40.8	10/13.2
	Oxidized polyvinyl alcohol hydrolase	* Syntrophorhabdus * sp. *PtaB*	A0A1V4WJI2	30.07	2.92 × 10^−10^	57	9.1/9.2/13.1
	Probable polyvinyl alcohol dehydrogenase	* Streptomyces rochei *	Q83X81	36.15	1.60 × 10^−6^	49.3	9.2/10/13.2
	PVA dehydrogenase PQQ dependent	* Bradyrhizobium * sp.	A0A160UKB5	28.36	4.92 × 10^−4^	41.6	9.2/10/13.2
**PHA** **PHB**	Poly(3-hydroxyalkanoate) depolymerase	* Pseudomonas fluorescens *	A0A0C1ZS59	100.00	0.00	577	9.2
	Poly(3-hydroxyalkanoate) depolymerase	* Pseudomonas putida S12 *	A0A0A7PVK5	99.65	0.00	574	10.13.2
	Poly(3-hydroxybutyrate) depolymerase	* Haladaptatus paucihalophilus *	E7QQJ1	50.73	2.83 × 10^−8^	55.5	9.1
	Poly(3-hydroxybutyrate) depolymerase	* Marinobacter lutaoensis *	A0A1V2DRR5	46.67	5.28 × 10^−24^	85.1	9.2
	Poly(3-hydroxyalkanoate) depolymerase C	* Paenibacillus polymyxa *	A0A2X1WPU8	44.53	4.46 × 10^−114^	338	9.1
	Poly(3-hydroxybutyrate) depolymerase	* Marinobacter * sp. *AC-23*	A0A1S2CI13	44.44	6.19 × 10^−17^	67.4	10/13.2
	Poly(3-hydroxyalkanoate) depolymerase	* Pseudomonas fluorescens strain Pf0-1 *	Q3KCH8	44.19	1.09 × 10^−4^	40.8	13.1
	Poly(3-hydroxybutyrate) depolymerase	* Bacillus megaterium ATCC12872 *	D5DZL2	43.26	2.37 × 10^−83^	251	13.1
	Poly(3-hydroxyalkanoate) synthase	* Paracoccus denitrificans *	Q9WX80	33.51	9.86 × 10^−98^	308	all

**Table 2 ijms-23-05612-t002:** Predicted plasticizer and other xenobiotic biodegradation genes with significant percentage identity and similarity to genes encoded in strains within the full consortium.

Enzyme	Species	Uniprot ID	% Identity	E-Value	Bit-Score	Strain
Taurine dioxygenase	* Gordonia phthalatica *	A0A0N9N9P8	68.09	3.71 × 10^−146^	410	9.2/10/13.2
2-nitropropanedioxygenase	* Gordonia phthalatica *	A0A0N9N4Y4	51.10	1.02 × 10^−83^	254	all
Tert-butyl alcohol monooxygenase reductase	* Aquincola tertiaricarbonis *	G8FRC6	44.59	2.50 × 10^−84^	255	all
4,4′-diaponeurosporenoateglycosyltransferase	* Bacillus enclensis *	A0A0V8HPX8	44.05	3.17 × 10^−10^	60.5	all
Phenol hydrolase reductase	* Methylibium petroleiphilum *	A2SI47	41.38	4.97 × 10^−11^	61.2	all
2-hydroxy-6-oxo-6-(2′-carboxyphenyl)-hexa-2,4-dienoate hydrolase	* Terrabacter * sp. *strain DBF63*	Q83ZF0	38.46	1.06 × 10^−18^	82.4	all
Tert-butyl alcohol monooxygenase	* Aquincola tertiaricarbonis *	G8FRC5	38.18	1.27 × 10^−4^	37.7	9.2/10/13.1/13.2
Quercetin 2,3-dioxygenase	* Gordonia phthalatica *	A0A0N9MT24	37.16	2.99 × 10^−34^	121	all
NidB2	* Mycobacterium vanbaalenii *	Q6H2J5	37.04	2.1 × 10^−10^	54.3	9.2/10/13.2
Naphthalene inducible dioxygenase	* Mycobacterium vanbaalenii *	Q9KHU1	35.64	3.51 × 10^−43^	155	9.2/10/13.1/13.2
5,5′-dehydrodivanillateO-demethylase	* Paraburkholderia tropica *	A0A1A5XFM6	34.38	4.84 × 10^−18^	82	all
Probable phenol hydrolase	* Rhodococcus * sp. *EsD8*	N1M644	33.33	7.2 × 10^−8^	51.6	all
Putative nitropropane dioxygenase	* Rhodococcus * sp. *DK17*	Q6REN2	32.84	2.46 × 10^−34^	127	all
1-hydroxy-2-naphthoicaciddioxygenase	* Mycobacterium * sp. *CH1*	C0KUL5	32.65	7.45 × 10^−58^	190	9.1
Phenol hydrolase	* Rhodococcus opacus M213 *	K8XRS6	29.667	3.2 × 10^−17^	79	all
2-3DHBA3,4-dioxygenase	* Pseudomonas stutzeri *	A0A2Z5UC95	29.524	3.65 × 10^−5^	42.4	10/13.2
Phenanthrene-4,5-dicarboxylate 5-decarboxylase	* Pseudonocardia * sp. *Ae707*	A0A1Q8KNT8	27.727	2.02 × 10^−9^	54.3	9.1/9.2/10/13.2

**Table 3 ijms-23-05612-t003:** Predicted phthalate plasticizer biodegradation genes with significant percentage identity and similarity to genes encoded in strains within the full consortium.

Enzyme	Species	Uniprot ID	% Identity	E-Value	Bit-Score	Strain
Phthalate 4,5-dioxygenase oxygenase reductase	* Pseudomonas * sp. *58R3*	A0A1B5EAD8	81.65	0.00	541	10/13.2
Phthalate 4,5-dioxygenase	* Pseudomonas fulva strain 12-X *	F6AJ53	64.67	1.82 × 10^−147^	416	10/13.2
Ferredoxin	* Burkholderia cepacia *	A0A1Z3YX76	57.05	2.73 × 10^−128^	367	9.2/10/13.2
Reductase component of isophthalate dioxygenase	* Comamonas * sp. *E6*	C4TNS5	54.05	2.47 × 10^−^^5^	43.1	all
Putative phthalate dioxygenase reductase	* Acinetobacter johnsonii SH046 *	D0SH70	51.43	4.11 × 10^−114^	330	10/13.2
Phthalate 4,5-dioxygenase	* Hydrogenophaga * sp. *PBC*	A0A1C9VA35	48.15	1.09 × 10^−5^	45.4	9.2
Phthalate-dioxygenase	* Hydrogenophaga intermedia *	A0A1L1P942	48.15	1.09 × 10^−5^	45.4	9.2/10/13.2
4,5-dihydroxyphthalatedecarboxylase	* Bacillus aquimaris *	A0A1J6W284	46.43	1.42 × 10^−11^	64.7	9.1
4,5-dihydroxyphthalatedecarboxylase	* Sporosarcina * sp. *P17b*	A0A2G5XE92	44.90	2.82 × 10^−5^	43.1	13.1
4,5-dihydroxyphthalatedecarboxylase	* Caballeronia megalochromosomata *	A0A149R8D0	44.00	7.82 × 10^−5^	41.6	9.2
Phthalate 4,5-dioxygenase oxygenase subunit	* Novosphingobium * sp. *MBES04*	A0A0S6WTD6	43.93	2.44 × 10^−82^	250	10/13.2
Phthalate 4,5-dioxygenase oxygenase reductase	* Bordetella pertussis H921 *	A0A0N2IN58	43.66	9.27 × 10^−7^	47.4	9.2
Putative phthalate dioxygenase reductase	* Bordetella pertussisH921 *	Q2YM46	43.66	9.27 × 10^−7^	47.4	9.2
Phthalate dioxygenase reductase	* Pandoraea sputorum *	A0A239SNB1	43.28	5.05 × 10^−8^	51.6	9.2
Aromatic ring-opening dioxygenase LigA	* Azoarcus * sp. *PA01*	A0A0M0FSY9	43.09	2.04 × 10^−46^	154	all
Extradiol ring-cleavage dioxygenase	* Gordonia phthalatica *	A0A0N9NE56	43.03	4.80 × 10^−51^	166	all
Putative phthalate dioxygenase reductase	* Brucella abortus strain 2308 *	Q2YM46	42.82	2.26 × 10^−102^	305	9.2
Phthalate 4,5-dioxygenase (Phthalate dioxygenase)	* Ramlibacter tataouinensis *	F5XWD6	41.96	5.06 × 10^−27^	109	all
Phthalate 4,5-dioxygenase oxygenase (OhpA2)	* Paraburkholderia xenovorans *	Q13QM0	41.82	8.89 × 10^−32^	122	all
Phthalate 4,5-dioxygenase	* Mycolicibacterium wolinskyi *	A0A1X2FHI8	40.74	4.99 × 10^−8^	52	9.1/13.1
Phthalate 4,5-dioxygenase oxygenase (OhpA2)	* Variovorax * sp. *WDL1*	A0A109CIC4	39.54	1.06 × 10^−4^	37.7	9.1/10/13.1/13.2
Phthalate 3,4-dioxygenase alpha subunit	* Klenkia soli *	A0A1H0Q6Z9	38.89	1.89 × 10^−11^	62.4	13.1
Phthalate 4,5-dioxygenase oxygenase reductase	* Gibberella fujikuroi *	A0A0I9YA52	38.73	2.29 × 10^−22^	92.4	9.1/13.1
Ferredoxin	* Brevirhabdus pacifica *	A0A1U7DHI8	38.21	1.30 × 10^−9^	57	9.1/13.1
Phthalate 3,4-dioxygenase alpha subunit	* Rhodococcus * sp. *OK302*	A0A235G3V7	38.18	2.65 × 10^−4^	38.5	9.1/13.1
Oxygenase large subunit of phthalate dioxygenase	* Terrabacter * sp. *strain DBF63*	Q8GI63	38.18	8.64 × 10^−4^	37	all
Phthalate 4,5-dioxygenase oxygenase subunit	* Thalassobius gelatinovorus *	A0A0P1FRT5	37.84	5.95 × 10^−20^	82.8	13.1
4,5-dihydroxyphthalate decarboxylase	* Pseudoruegeria lutimaris *	A0A1G9AEN9	37.50	9.18 × 10^−5^	41.6	10/13.2
Phthalate dioxygenase reductase	* Gibberella subglutinans *	A0A109QSZ1	37.27	1.64 × 10^−26^	104	9.1
Phthalate 4,5-dioxygenase oxygenase subunit	* Alphaproteobacteria bacterium *	A0A2S6QA16	36.84	4.36 × 10^−4^	35.8	9.1
Phthalate 3,4-dioxygenase alpha subunit	* Mycolicibacterium rutilum *	A0A1H6J828	35.48	7.71 × 10^−6^	45.4	9.2
Oxygenase component of isophthalate dioxygenase	* Comamonas * sp. *E6*	C4TNS2	34.43	1.37 × 10^−28^	113	9.2/10/13.1/13.2
Putative phthalate dioxygenase reductase	* Providenciaal califaciens PAL-3 *	W3YHJ5	33.64	6.31 × 10^−9^	54.7	9.1/13.1
3,4-dihydroxyphthalatedecarboxylase	* Arthrobacter * sp. *strain FB24*	A0AWN5	33.49	2.26 × 10^−17^	75.9	9.1/13.1
3,4-dihydroxy-3,4-dihydrophthalate dehydrogenase	* Terrabacter * sp. *strain DBF63*	Q8GI60	33.15	2.80 × 10^−8^	51.6	all
3,4-dihydroxyphthalatedecarboxylase	*Nocardioides terrae*	A0A1I1EG61	33.15	3.34 × 10^−17^	76.6	10/13.2
3,4-dihydroxyphthalatedecarboxylase	* Klenkia soli *	A0A1H0Q8Z1	32.52	1.20 × 10^−14^	69.3	9.2
Phthalate-dioxygenase	* Bradyrhizobium * sp. *ORS3257*	A0A2U3Q6T0	32.22	1.48 × 10^−23^	97.8	13.1
Cis-phthalate dihydrodiol dehydrogenase	* Comamonas * sp. *E6*	A0A0M2DHI3	31.90	2.50 × 10^−18^	82.8	9.2
Phthalate 3,4-dioxygenase alpha subunit	* Rhodococcus rhodnii LMG5362 *	R7WIP7	31.63	1.31 × 10^−38^	140	10/13.2
Cis-phthalate dihydrodiol dehydrogenase	* Burkholderia multivorans *	A0A0H3KKN4	28.89	2.13 × 10^−8^	52.8	9.1/10/13.1/13.2

**Table 4 ijms-23-05612-t004:** Differential expression of upregulated genes within the individual strains that may be involved in PET degradation.

Strain	Gene	logFC	logCPM	*p*-Value
9.1	Poly(3-hydroxyalkanoate) depolymerase	3.42	2.09	3.64 × 10^−2^
	Phthalate 4,5-dioxygenase, reductase subunit	2.60	8.24	2.20 × 10^−9^
	Short-chain dehydrogenase/reductase	1.90	3.59	6.32 × 10^−3^
	Alcohol dehydrogenase 2	1.47	3.50	3.79 × 10^−2^
	Poly(3-hydroxyalkanoate) depolymerase	1.35	10.69	4.09 × 10^−3^
	Phenol hydrolase	1.31	10.94	1.68 × 10^−4^
	Poly(3-hydroxybutyrate) depolymerase	1.14	12.18	8.31 × 10^−4^
	Glyoxal reductase	0.89	7.14	1.58 × 10^−2^
	Gentisate transporter	0.70	9.60	4.30 × 10^−2^
9.2	Taurine dioxygenase	3.67	2.06	4.91 × 10^−3^
	Putative regulatory protein	2.90	2.32	1.75 × 10^−2^
	Short-chain dehydrogenase/reductase	1.21	5.19	2.51 × 10^−2^
	Beta-carboxy-cis,cis-muconate cycloisomerase	1.05	10.95	2.52 × 10^−3^
	Beta-ketoadipyl CoA thiolase	0.97	8.61	1.10 × 10^−2^
	Aldehyde dehydrogenase	0.89	8.35	3.30 × 10^−2^
	Poly(3-hydroxyalkanoate) depolymerase	0.78	9.45	3.32 × 10^−2^
10	Poly(3-hydroxybutyrate) depolymerase	2.80	−0.54	3.93 × 10^−3^
	3,4-dihydroxyphthalate decarboxylase	2.48	7.72	2.19 × 10^−8^
	Terephthalate 1,2-dioxygenase, oxygenase	1.32	1.20	3.09 × 10^−2^
	Putative regulatory protein	1.19	6.38	4.34 × 10^−4^
	Putative regulatory protein	1.09	5.72	9.15 × 10^−4^
	4,5-dihydroxyphthalate decarboxylase	1.07	5.63	8.44 × 10^−4^
	Poly(3-hydroxybutyrate) depolymerase (*nlhH*)	1.04	5.15	2.35 × 10^−3^
	Beta-carboxy-cis,cis-muconate cycloisomerase	0.84	9.74	7.32 × 10^−3^
	Putative regulatory protein	0.78	9.89	3.17 × 10^−2^
	Poly(3-hydroxybutyrate) depolymerase	0.74	10.89	1.74 × 10^−2^
	Quercetin 2,3-dioxygenase	0.68	6.18	3.37 × 10^−2^
	Surfactin synthase subunit 3	1.03	4.21	4.71 × 10^−3^
13.2	Poly(3-hydroxybutyrate) depolymerase	2.80	−0.54	3.89 × 10^−3^
	3,4-dihydroxyphthalate decarboxylase	2.48	7.72	2.11 × 10^−8^
	Terephthalate 1,2-dioxygenase, terminal oxygenase	1.32	1.20	3.09 × 10^−2^
	Putative regulatory protein	1.23	5.24	2.26 × 10^−3^
	Putative regulatory protein	1.19	6.38	4.31 × 10^−4^
	Poly(3-hydroxybutyrate) depolymerase (*nlhH*)	1.13	5.11	8.26 × 10^−4^
	Beta-carboxy-cis,cis-muconate cycloisomerase	0.84	9.74	7.31 × 10^−3^
	Putative regulatory protein	0.78	9.89	3.14 × 10^−2^
	Poly(3-hydroxybutyrate) depolymerase	0.74	10.90	1.73 × 10^−2^
	Poly(3-hydroxyalkanoate) depolymerase	0.73	8.09	2.62 × 10^−2^
	Quercetin 2,3-dioxygenase	0.68	6.18	3.32 × 10^−2^

## Data Availability

RNAseq datasets were deposited in the NCBI’s Sequence Read Archive (SRA) and are publicly available on GenBank, BioProject Accession PRJNA517285.

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
