# Peer review of "Microbial Consortia and Mixed Plastic Waste: Pangenomic Analysis Reveals Potential for Degradation of Multiple Plastic Types via Previously Identified PET Degrading Bacteria"

_ijms, 2022, doi:10.3390/ijms23105612_

Round 1

Reviewer 1 Report

The paper is prepared very well and correctly. No significant problems have been seen. it can be accepted for publishing.

 This is a hot topic. While the development of highly degradable bacteria of recycled plastics is an actual area of science itself and is of great importance for various applications of plastics. The work is interesting for providing methods for choosing the strain for degradation control and in summary to develop a methodology for the process. The discussion is clear, the selection of materials well justified and the analysis novel enough to merit publication.
Indeed, the suggested manuscript combines a comprehensive fundamental study and applied investigation. The work is interesting and well done, making a good impression from the article.
I have no major criticisms or suggestions to significantly modify the work done.

Author Response

Dear Reviewer #1,

 Thank you for taking the time to thoroughly read the manuscript. 

Sincerely,

Jay Mellies, Ph.D.

Amgen/Perlmutter Professor of Biology 

Biology Department

Reed College

3203 SE Woodstock Blvd

Portland, OR 97202

email: jay.mellies@reed.edu

Office: 503.517.7964

Reviewer 2 Report

Reviewer comments:

The presentation of this manuscript was good enough in terms of structure and English language. Moreover, the paper is subjected to some improvement before being accepted for publication; specially in discussions.

  1. The problem statement with identification of the novelty of this work have to state clearly.
  2. The condition that enhance or stimulate the enzymatic activities the was responsible for plastic degradation need to define properly.
  3. More degradation (plastics) evaluation are needed.   

Author Response

Dear Reviewer #2,

First, we thank you for taking the time to carefully read our manuscript submitted to IJMS, and we appreciate the comments aimed toward improving the submission. Punctuation and spelling errors have been corrected. We have addressed the comments below, edited the work accordingly, and have provided an updated manuscript with Track Changes.

Response:

  1. The problem statement with identification of the novelty of this work have to state clearly.

To our knowledge, this is the first report of combining pangenomics and transcriptomics analysis of a bacterial consortium to elucidate PET cleavage to complete mineralization of the polymer. We also use molecular approaches, gene knockout and protein purification, with biochemical assays to support the identification of the novel PETase EstB. The statement of the problem, along with these novelties of the work are included in the Conclusions beginning on line 767. While the reviewer suggested that this comment should be addressed in the Discussion, the authors saw that the information was better suited to the Conclusions.

  1. The condition that enhance or stimulate the enzymatic activities the was responsible for plastic degradation need to define properly.

For this report, we describe the PETase EstB’s ability to degrade micro-particles of PET plastic. The methods for production of the micro particles are described in Materials and Methods beginning on line 721. Because the crystallinity of PET water bottles (~30%) inhibits the enzymatic cleavage by all PETases described to date, we used micro-PET particles to increase the exposed surface area to enhance degradation.

Line 721. “MicroPET was synthesized as outlined by Rodríguez-Hernández et al.85 in order to increase the exposed surface area of the material and to enhance degradation.”

  1. More degradation (plastics) evaluation are needed.

In this report, we provide experimental evidence that purified EstB, a novel PETase, degrades BHET and PET to form the monomers MHET, TPA and ethylene glycol (see Figure 3). However, the focus of the submission, and indeed the special issue is "Biodegradation of Pollutants in the Environment: Omics Approaches.” Using pangenomics and transcriptomics we have identified over 200 enzymes that are predicted to, but not demonstrated to degrade multiple types of man-made plastics, bioplastics as well as biopolymers. Thus, we concluded that follow-up experiments will be needed to further explore the degradation capabilities of the consortium of bacteria but this effort is beyond the scope of the current submission. However, it is an approach that our research group will certainly pursue.     

We hope that the manuscript is now acceptable for publication, and we look forward to your response.

Sincerely,

Jay Mellies, Ph.D.

Amgen/Perlmutter Professor of Biology

Biology Department

Reed College

email: jay.mellies@reed.edu